# Yoga as an Integrative Therapy for Mental Health Concerns: An Overview of Current Research Evidence

**Crystal L. Park** [1,*]  **and Jeanne M. Slattery** [2]

[1] Department of Psychological Sciences, University of Connecticut, Storrs, CT 06269, USA
[2] Department of Psychology, Clarion University of Pennsylvania, 840 Wood St, Clarion, PA 16214, USA; jslattery@clarion.edu
* Correspondence: crystal.park@uconn.edu

**Abstract:** Background: Because the prevalence of mental health concerns is high and access or full responsiveness to pharmacological or psychotherapeutic treatment for many individuals is low, there has been increased interest in yoga as a potential therapy for many mental health concerns. Approach: We synthesize and critique current research on the efficacy of yoga relative to pharmacological approaches for anxiety disorders, mood disorders, posttraumatic stress disorder, obsessive-compulsive disorder, and eating disorders. Results: Yoga has been tested mostly as a complementary treatment to standard psychiatric and psychotherapeutic approaches. Findings from efficacy trials largely support the notion that yoga can help reduce symptoms of many psychiatric conditions, including anxiety, depression, and PTSD symptoms, above and beyond the effects achieved by standard pharmacological treatments alone; however, most evidence is of poor to moderate quality. Plausible transdiagnostic bottom-up and top-down mechanisms of yoga's therapeutic effects have been advanced but remain untested. Conclusions: While results should be considered preliminary until more rigorous evidence is available, yoga appears to have the potential to provide many people suffering with psychiatric symptoms additional relief at relatively little cost. Yoga may be a viable complementary therapy to psychiatric and psychotherapeutic approaches for people with mental health challenges.

**Keywords:** yoga; anxiety disorders; depressive disorders; PTSD; eating disorders



## 1. Introduction

Mental health is an increasingly urgent global public health concern. Untreated, mental illnesses account for a large share of global disease burden, estimated at 7 to 19% of the total [1]. Within 20 years, depression alone is expected to be the leading cause of global disease burden [2]. The costs in terms of suffering and lost potential to individuals, families, and society is immense. In spite of recent advances in technology, evidence-based treatments, and care delivery, many people suffering from mental illnesses remain untreated or receive only marginally effective treatments. Given that full alleviation of suffering from mental illness is uncommon, additional approaches are needed to complement these treatments, extend efficacy, and offer options acceptable to people who desire treatments other than or in addition to medication or psychotherapy.

One recently emerging approach to mental health is drawn from a very old Eastern tradition: Yoga. Yoga is a broad umbrella term for a variety of philosophical tenets and lifestyle practices, the most common of which in biomedical cultures is hatha yoga. Hatha yoga comprises physical postures, regulation of breathing, and meditation (see [3] for an overview).

While hatha yoga is often practiced as a form of physical exercise, it is now frequently studied as a treatment for a variety of mental and physical health conditions. The prevalence of yoga practice has increased significantly in recent years. In the US, for example, approximately 13.2% of adults report having practiced yoga, almost 163% more in 2012 than in 1998 [4]. Many individuals report practicing yoga to improve their mental health,

reduce stress, or bolster their mood [5]. Because many people already practice yoga to manage their mental health conditions and because many others may benefit, mental health professionals may want to be conversant with this expanding body of research and may consider integrating yoga into their treatment approaches.

In biomedicine, yoga is usually considered a complementary treatment; that is, a practice that can be integrated into a larger treatment plan that is based primarily on pharmacological treatment or psychotherapy. Earlier, yoga was sometimes considered part of "alternative" medicine, a broad body of treatments that attempt to provide alternatives to standard biomedicine. In the past few decades, many so-called alternative medical modalities have been subjected to scientific scrutiny to determine their value in treating a variety of health conditions. Those shown in clinical research to be of some value are often integrated into standard psychiatric and psychotherapeutic approaches to provide additional potential therapeutic benefit.

Because yoga is widely considered a way to promote mental health in the broader culture, clinical studies of yoga have proliferated in the past few decades, resulting in a large literature base. Many subsequent reviews and meta-analyses have been conducted in efforts to integrate this work (cf. [6–10]); however, making sense of these reviews and meta-analyses is complicated and difficult. The body of studies on which they draw is riddled with poor-quality clinical trials that fail to consider or eliminate bias. Further, definitions of "yoga" vary widely; thus, these published reviews of yoga have often included any intervention labeled as yoga by the authors, including Sudarshan Kriya yoga, which is almost entirely pranayama-based (breathwork), and Sahaj yoga, which is primarily seated meditation (cf. [7–9]). Most Westerners recognize yoga as including both of these aspects but also see asanas (yoga postures) as essential to yoga practice.

To increase our understanding of the efficacy of yoga for mental health conditions, we provide an overview of the broad body of empirical literature, focusing specifically on studies of asana-based yoga, which often also includes elements of meditation and breathwork. Extant published reviews and meta-analyses of yoga's efficacy for people with mental health diagnoses have generally not considered the other treatments in which patients or participants may be engaged (e.g., psychiatric medicine and psychotherapy). To better contextualize the efficacy of yoga with more mainstream treatments, we organized our overview according to whether the studies tested yoga as an alternative to psychopharmacological or other first-line treatments or as an integrative or complementary treatment used along with standard treatments. We reviewed studies on hatha yoga's efficacy for five of the most common categories of mental illness: anxiety disorders, depression, obsessive–compulsive disorder, PTSD, and eating disorders. Although we did not conduct a systematic review, we endeavored to include all published studies conducted with reasonable rigor that addressed the issue of efficacy with these five categories of disorders.

## 2. Methods

To identify studies for this overview, we relied on a set of recently-published meta-analytic reviews of each category of mental health condition [6–10]. We considered each of the studies cited within each review for inclusion and identified more recent studies through searches of PubMed, PsycINFO, and Google Scholar using as the search terms "yoga" and multiple variants of each of the diagnostic categories included here. Both authors examined each study to determine their eligibility for inclusion. The retained studies were those that had tested the efficacy of hatha yoga specifically, rather than only described its use, and that included information on other treatments provided and the nature of those treatments. We excluded studies that focused on pranayama- and meditation-only styles of yoga (e.g., Sudarshan Kriya yoga and Sahaj yoga) if they did not also include asanas as part of the yoga practice. Additionally, studies were not included if the articles weakly communicated designs or outcomes and prevented comparison or analysis of outcomes. These criteria meant we omitted some studies included in the previously cited meta-analyses.

In our overview, we distinguished between studies that consider yoga as a primary (alternative) or adjunctive (complementary or integrative treatment) treatment and have endeavored to describe the role of other first-line treatments when this information was reported.

## 3. Anxiety Disorders

Anxiety disorders are characterized by feelings of worry, anxiety, dread, or fear that are strong enough to interfere with a person's performance at work or school, in relationships, or in other important domains. They are associated with autonomic arousal, thoughts of impending danger, and avoidance or escape behavior [11]. Anxiety becomes problematic when it is out of proportion to the situation or developmental stage, persistent beyond the instigating stressor, and hinders functioning. Worldwide, prevalence estimates for anxiety disorders are around 7.3%, ranging from 4.8% to 10.9% [12]. Diagnosed anxiety disorders include generalized anxiety disorder, phobia, agoraphobia, panic disorder, social anxiety disorder, and separation anxiety disorder.

First-line treatments for high levels of anxiety and anxiety disorders include psychotherapeutic approaches, especially cognitive–behavioral therapy (CBT), and medication [13]. Patients who receive psychiatric medications often experience suboptimal anxiety reduction. For example, satisfactory responses to short-term selective serotonin reuptake inhibitors (SSRIs) and serotonin–norepinephrine reuptake inhibitors (SNRIs) for the treatment of generalized anxiety disorder (GAD) occur in approximately 60% of patients [14]. Approximately 30% of patients recover with standard pharmacology treatments, while another 30–40% are considered improved [15]. Relapse rates are very high and discontinuing medication greatly increases the odds of relapse [16].

Many people reporting high levels of anxiety do not seek medical treatment and reject psychotherapeutic or psychiatric interventions [17]. Concerns regarding potential side effects, high costs of treatment, potential addiction to medication, and dissatisfaction with pharmacological treatments contribute to low treatment adoption and adherence rates [18,19].

People suffering from anxiety disorders often choose to manage their anxiety on their own rather than by seeking medical treatment or engaging in psychotherapy or psychiatric interventions [20]. They often use self-help approaches such as support groups and physical exercise [20] and may also rely on alcohol or other psychoactive substances to alleviate their symptoms [21].

The appeal of yoga as an additional method to attempt to relieve anxiety has been growing in recent years. Indeed, reducing stress and anxiety is a primary reason people give for practicing yoga [5]. In the context of anxiety, yoga intervention research has occasionally considered yoga as an alternative to standard treatment but has almost always taken a complementary approach, offering yoga in addition to whatever treatment individuals were already receiving for their anxiety.

### 3.1. Yoga as an Alternative Approach for Anxiety Disorders

A few studies have directly compared yoga to pharmacotherapy. In an early study, 91 patients diagnosed as DSM-III "anxiety–neurotic" at a psychiatric outpatient clinic in India were randomized to receive yoga for 40 min/day five days/week for three months or given diazepam. Pre–post scores on an anxiety measures significantly improved for the yoga group and did not significantly change for the medication group [22].

Some studies have directly compared yoga with standard psychotherapeutic treatments while psychiatric medication management was administered across conditions. A recent study of 226 adult patients (mean age of 33, 70% female; 6% were on anxiolytics) with a primary DSM-5 GAD diagnosis were randomized to a 12-week trial of Kundalini yoga, CBT, or stress education [23]. The patients in the Kundalini yoga condition showed significant pre-to-post trial improvements in anxiety relative to the stress education group, although the CBT group improved even more. The authors concluded that Kundalini

yoga was efficacious for GAD, although the results support CBT as first-line treatment. A small study of 20 patients diagnosed with panic disorder were recruited from a Brazilian psychology clinic and randomized to receive either yoga or a combination of yoga and psychotherapy [24]. No information on medication was provided. Both interventions occurred weekly for 100 min and lasted 2 months. Significant reductions in anxiety levels associated with panic disorder, panic-related beliefs, and panic-related body sensations were observed in both treatment arms; however, the combined yoga and CBT group showed even further reductions in anxiety.

*3.2. Yoga as a Complementary or Integrative Approach for Anxiety Disorders*

In most studies of yoga as a treatment for anxiety, yoga is implemented as an adjunct to psychotherapy or medical treatment. A handful of these studies were conducted with people with diagnosed anxiety disorders. In one single-arm study, 55 adults in an acute inpatient psychiatric unit (with a variety of psychiatric diagnoses) participated in 1–2 yoga sessions in addition to treatment as usual (TAU) [25]. Patients who completed at least one session of yoga reported reductions in anxiety that lasted up to a full day; participants also reported using yoga and meditation as coping mechanisms at discharge. A recent study in India recruited 200 patients with anxiety-related disorders from a hospital psychiatry clinic [26]. Patients were randomized into a 3 month yoga or relaxation intervention; the yoga group demonstrated substantial reductions in anxiety relative to the relaxation group from baseline to end of intervention. Another study tested the efficacy of two different interventions and a non-interventional control group in mitigating the effects of depression or anxiety [27]. Ninety college students diagnosed with anxiety or depression (58.20% diagnosed with both; 31.34% anxiety only, 10.44% depression only; 56.6% were taking psychiatric medications) were randomized into an 8 week mindfulness meditation intervention group, an 8 week yoga group, or a non-interventional control group. Anxiety symptoms decreased significantly from baseline to end of intervention in both the mindfulness and yoga intervention groups relative to the control group.

Other studies focused on non-patient community samples who were recruited on the basis of reporting elevated levels of anxiety. For example, a clinical trial conducted in Australia included 101 people with at least mild anxiety or depression as reported on the DASS-21, a symptom measure [28]. Participants were randomized to receive 6 weeks of individualized yoga or be on a waitlist. Significant differences in anxiety and distress symptoms were noted from pre-to-post assessment in the yoga group, while anxiety and distress levels of those on the waitlist control did not change. A randomized preference trial was conducted with 500 community-dwelling older adults (mean age was 66 years, 87% women, 44% were on psychotropic medications), who scored high on a worry questionnaire [29]. Participants received 10 weekly sessions of CBT or 20 twice-weekly group yoga classes. Both groups experienced substantial reductions in worry and anxiety that did not differ between groups. Preference for yoga or CBT did not influence results.

Another randomized controlled trial was implemented in Australia with 131 community residents who reported experiencing mild to moderate levels of stress. Twenty-four percent of the sample was taking psychotropic medication [30]. Participants received 10 weekly sessions of relaxation or hatha yoga. Pre-to-post intervention scores on anxiety improved over time, but improvement did not differ by intervention.

Several studies have been conducted with medical patients with elevated anxiety. One randomized clinical trial of 38 breast cancer outpatients undergoing conventional treatment at a cancer center compared the anxiolytic effects of a yoga program to supportive therapy prior to their primary treatment (i.e., surgery) [31]. About 40% were taking anxiolytics. The results showed significant decreases in anxiety pre-to-post in the yoga group relative to supportive therapy. Another non-psychiatric medical-patient-focused study examined 250 people with osteoarthritis who had received transcutaneous electrical stimulation and ultrasound treatment [32]. They were randomly assigned to yoga or a TAU control

group for three months. Anxiety decreased in both groups but substantially more so in the yoga intervention.

Yoga was also studied in a healthy community sample with "normal" levels of anxiety. In this study, 34 healthy community participants with no noted psychiatric disorders were randomized to either 12 weeks of 3/week Iyengar yoga or walking. Those in the yoga group experienced greater reductions in anxiety relative to those in the walking group [33]. Participants were allowed to be on psychotropic medications although no information on medication use was reported. A study from India randomized women referred to a yoga clinic to either a yoga (twice weekly, 90 min duration, for two months) or a waitlist control group [34]. Both groups were evaluated at baseline and end of the two-month intervention. Women participating in yoga classes reported significant decreases in both state and trait anxiety, while women's anxiety levels in the control condition did not change.

*3.3. Summary of Yoga's Anxiolytic Effects*

In summary, yoga demonstrated substantial effects on anxiety in nearly all of the studies reviewed, which cover a broad range of samples, types of yoga, and study designs. Although some of the studies focused on diagnosed anxiety disorders, the majority were conducted with people who self-identified as having high levels of anxiety symptoms. The fact that in spite of this heterogeneity, yoga interventions were almost always efficacious suggests a robust effect.

Yoga's anxiolytic effects may be due to a number of different physiological, psychological, and behavioral mechanisms. Yoga may promote emotion regulation through integrating bottom-up physiological and top-down psychological processes that facilitate bidirectional communication between mind and body [35]. Yogic breathing and movement enhance autonomic nervous system (ANS) regulation. Through practicing yoga, individuals develop skills in remaining calm in times of challenge through deep breathing, mindful awareness, and attention. Yoga practices can stimulate the vagus nerve, helping to increase the balance of the ANS through the proportions of GABA and glutamate [36]. Also of importance, practicing yoga facilitates autonomic balance by increasing heart rate variability (HRV). Increased HRV is also associated with improved adaptation to changing environmental stimuli and physiological reactions to stress [37–39], while higher HRV promotes recovery following stressful situations [40]. Other potential top-down processes include attention control, emotional balance, coping abilities, and perspective and wisdom [35].

Although these studies are fairly consistent in demonstrating that yoga as an adjunct to standard first-line therapies may bring about additional relief from anxiety, caution must be taken in drawing firm conclusions. The studies reviewed were generally of fairly weak design, often using waitlist or TAU controls, which do not permit separating the effects of yoga from non-specific factors such as expectancies and attention. Further, the clinical trials reported here are prone to many types of biases that are pervasive in nearly all clinical trials [41]; thus, yoga may be considered a complementary option in treating a range of anxiety disorders given its potential effectiveness and minimal disadvantages, although stronger clinical trials are needed to build a firmer evidence base.

## 4. Major Depressive Disorder and Depressive Symptoms

Major depressive disorder (MDD) is characterized by changes in mood (depressed mood), motivation (loss of interest or pleasure in daily activities), cognition (inappropriate guilt, diminished concentration, and recurrent thoughts of death or death-furthering thoughts and behaviors), and physical symptoms (increases or decreases in eating, sleeping, activity, or energy), which significantly affect a person's ability to function [11]. In the US, the 12 month prevalence for MDD is 7%; women are more likely to be diagnosed with MDD than men, and young people (18–29 years) are three times as likely to be diagnosed as people who are 60 or older [11].

Recommendations for treating MDD often include antidepressants and cognitive and interpersonal therapies. Antidepressants have a somewhat more rapid effect for people

with moderate to severe MDD and is equivalent in efficacy to cognitive therapy by 16 weeks; people receiving cognitive therapy are less likely to relapse following treatment [42]. Several non-pharmacological treatments have also been shown to be efficacious, including aerobic exercise [43], behavioral activation [44], and mindfulness-based interventions [45]. Yoga may also be an effective treatment for MDD (cf. [7]).

### 4.1. Yoga as a Complementary or Integrative Approach for MDD and Depressive Symptoms

Studies of yoga as an integrative and complementary treatment compare participants receiving treatment as usual (TAU) plus yoga to TAU alone. In this case, participants may receive an antidepressant, cognitive therapy, or other treatments. For example, one study randomly assigned participants in a psychiatric hospital diagnosed with an affective disorder (42.5% with MDD) to either TAU or TAU plus a 30-day trial of asanas, breathwork, and meditation [46]. Members of the yoga group reported a significantly greater decrease in depression and anxiety scores and significantly greater clinical improvement relative to the control group over the trial. Another clinical trial assigned people with MDD to one of three groups: an 8-week Bikram group, aerobic exercise, or a waitlist control [47]. Remission rates in the yoga and aerobic exercise groups were similar at 61.1% and 60.0%, respectively, of the levels at the start of treatment and were significantly higher than waitlist controls (6.7%). Reductions in depressive symptoms in both active groups were mediated by changes in rumination. About 20% of their full sample was on medication and about 20% were receiving psychotherapy.

On the other hand, a study compared an inpatient population receiving one of two antidepressants or an antidepressant plus a 5 week course of weekly hatha yoga in a randomized trial [48]. Although both groups (yoga plus TAU vs. TAU) reported a significant decrease in symptoms over time, the yoga plus TAU group did not report a significantly greater decrease in symptoms. Similarly, comparable decreases in depressive symptoms were reported in a study of women with diagnosed MDD or dysthymia who were randomly assigned to either an eight week weekly hatha yoga group with home practice or an attention control group [49]. Nearly 70% of participants reported being in psychotherapy. Almost two-thirds reported taking an antidepressant; somewhat less than 40% were taking an anxiolytic. Those in the yoga plus TAU group reported significantly lower rumination scores at the end of the trial than did participants in the attention-control plus TAU group.

Some intervention studies did not explicitly compare yoga to other treatments, but rather only compared doses of yoga [50]. For example, one trial assigned people diagnosed with MDD to receive either a high or low dose of home and studio Iyengar yoga (90 min class, three vs. two times per week for 12 weeks); 87% of the high dose group and 73% of the low dose group responded with more than a 50% decrease in scores on the Beck Depression Inventory-II (BDI-II), while both groups experienced remission at the same rate (BDI-II scores less than 14 at week 12: 93% of high dose group, 87% of low dose). In another study from the same research group, participants diagnosed with MDD were randomly assigned to either a low or high dose of home and studio Iyengar yoga, as above [51]. Eight of the nine participants who had been suicidal at the start of the study no longer reported suicidality at the end of the intervention. Although their sample size was small, the reported change did not appear to be dose-dependent. Note that the "low dose" of yoga described in these studies was more intense and longer in duration than that described in most other studies reported in the present review.

Other studies have directly compared participants with MDD receiving a trial of yoga to participants receiving other treatments. For example, adults with untreated MDD were randomly assigned to either an eight-week, twice-weekly hatha yoga class or an attention-control education group [52]. Yoga participants reported significantly greater changes in depressive symptoms and were more likely to experience remission from MDD during this period; however, they did not report changes in self-esteem or self-efficacy. Another study randomly assigned women with a diagnosis of MDD to either a 12-week mindfulness-based yoga intervention (home-based yoga asana, breathwork, and meditation practice with

telephone-delivered mindfulness education sessions) or a walking condition (home-based walking sessions and telephone-delivered health education sessions) [53]. Both groups reported similar and significant decreases in depressive symptoms over the course of the study, although the yoga group reported a significant decrease in rumination symptoms relative to the walking group. Another study randomly assigned women with diagnoses of MMD or dysthymia to either a yoga intervention or a health education group [54]. Both groups met weekly for 8 weeks for 75 min/week. The yoga group reported significant decreases in symptoms of depression and rumination relative to the health education group, and these decreases were maintained over a year; however, the one year follow-up results are difficult to interpret due to high attrition from the study (only 7 of 15 members of the yoga group and 2 of 12 of the health education group completed the follow-up assessment).

Yoga has also been tested with women diagnosed with MDD during pregnancy. This is an especially important intervention target, as many pregnant women prefer non-pharmacological treatments for their depressive symptoms. In one group, 92 women diagnosed with prenatal depression were randomly assigned to a hatha yoga group designed for women during their second and third trimester (20 min per week for 12 weeks) or a leaderless social support group [55]. The yoga did not include either breathwork or meditation. At the end of the trial, both groups reported significantly less depression, anxiety, and anger and improved relationships, and the effects were similar across the two groups. Changes in depression and anxiety were maintained at their follow-up postpartum. In another study, 46 pregnant women with elevated anxiety or depression were randomly assigned to eight weeks of prenatal yoga (8 weeks of 75 min sessions weekly) or TAU, which was accessed outside of the study [56]. Yoga was perceived as safe, feasible, and acceptable. Depression symptoms in the yoga group did not differ significantly from those reported by the TAU group, although the yoga group experienced significantly greater decreases in negative effects. Some members of the TAU and yoga groups used an antidepressant or received psychotherapy during the study period and four members of the TAU group performed yoga on their own.

### 4.2. Summary of Yoga for MDD and Depressive Symptoms

Multiple studies suggest that yoga interventions reduce the psychological and physical symptoms of depression in populations with clinical levels of depression and in those with subclinical symptoms (cf. [7,57–59]). Participants with depression find yoga interventions acceptable and beneficial (cf. [49,54,56]). In their meta-analysis, Haller et al. [57] concluded that yoga's effect on treating MDD was of a "large size" relative to TAU and "medium size" relative to standard interventions. This conclusion was consistent with that of a review by Cramer, Anheyer, Lauche, and Dobos [7]. Our review includes more articles than did the review by Cramer et al. [7], but also concludes that yoga is a promising intervention for MDD. We found that, relative to an active control, two of the four study groups performed significantly better than the control group [52,54]. When compared with active controls (e.g., walking or social support), they performed as well [53,55,56]. Both studies that looked at changes in rumination reported significant decreases in symptoms [53,54]. Studies of complementary interventions (interventions plus TAU) were more mixed in outcomes. Some supported the use of yoga as a treatment for MDD [46,47]; others did not see any additional advantage compared to TAU or an attention control [48,49].

Nonetheless, similarly to Haller and colleagues [57], we conclude that the data in these studies were often of very low quality, as there were often very low doses of yoga, no active control groups, considerable dropouts from the study, insufficient blinding of participants and researchers, and other biasing factors.

## 5. Obsessive–Compulsive Disorder

Obsessive–compulsive disorder (OCD), although not technically classified as an anxiety disorder, can be a life-long disorder often featuring high levels of anxiety and psychosocial impairment. SSRIs are the primary pharmacological approach, but approximately

50% of patients do not respond to this treatment. When combined with the standard behavioral therapy for OCD, exposure and prevention, about 30% of OCD patients remain non-responders [60].

We found just one study that examined yoga and OCD. A randomized clinical trial compared Kundalini yoga to a relaxation response intervention in 48 patients who met criteria for OCD; 52% were taking psychiatric medication [61]. Both groups improved in OCD symptoms, although the yoga group demonstrated greater improvements than did the relaxation response group, suggesting that Kundalini yoga may be considered an effective add-on for OCD patients unresponsive to more traditionally used treatments.

## 6. Posttraumatic Stress Disorder

Many people experience posttraumatic stress symptoms after experiencing or witnessing a traumatic event such as a natural disaster, a serious accident, a terrorist act, combat, or rape, or after being threatened with death, sexual violence, or serious injury [62]. These symptoms include intrusive thoughts of the trauma, avoidance, hyperarousal, and disturbances in cognition and mood [11]. When symptoms are severe and last at least one month, a diagnosis of posttraumatic stress disorder (PTSD) is applied. The annual prevalence of PTSD in the US is 3.5% and the lifetime prevalence is 9% [62]. Women are twice as likely as men to be diagnosed with PTSD [63].

Psychiatric treatment of PTSD typically consists of antidepressants such as SSRIs and SNRIs, which are used either alone or in combination with psychotherapy or other treatments. Recent meta-analyses reported small differences in outcomes between most pharmacological treatments for PTSD and placebos; nonetheless, medication may be helpful in controlling symptoms of PTSD, which may in turn help those with PTSD to engage in psychotherapy more effectively [64,65].

Psychotherapy, especially trauma-specific therapies such as prolonged exposure therapy or cognitive processing therapy, appear to be superior to medication as first-line treatments for PTSD [66]. These therapies typically focus on extinguishing conditioned fear responses, requiring patients to manage intense emotions while focusing on conditioned stimuli, such as sensations from the environment or one's memories [67]. Rates of premature termination from psychotherapeutic treatments for PTSD can be high. These high rates of attrition have been attributed to difficulties that many patients experience in tolerating these treatments [67].

Given these treatment difficulties, complementary therapy approaches for individuals with diagnosed PTSD or high levels of posttraumatic stress symptoms have received increasing interest by both mental healthcare providers and patients themselves. In particular, mind–body approaches may decrease trauma-related symptoms and improve emotion regulation [35], meaning they could help patients tolerate psychotherapy. Only a small number of studies have yet examined the efficacy of yoga for treating PTSD; we were unable to identify any articles directly comparing yoga to psychopharmacological approaches. All of the reviewed studies either allowed participants to continue with their other treatments as usual or did not mention other treatments at all in their published articles.

### 6.1. Yoga as a Complementary or Integrative Therapy for PTSD

In one of the earliest studies of yoga for PTSD, 64 women diagnosed with chronic, treatment-resistant PTSD were randomly assigned to either a trauma-informed yoga or supportive women's health education group. Each intervention took place for one-hour weekly over 10 weeks [68]. Study participants were required to be engaged in ongoing supportive therapy and continue current pharmacologic treatment. Women in both conditions showed decreases in PTSD symptoms by the end of the intervention, although the decreases in the yoga group were much larger.

Several yoga clinical trials have been conducted specifically with veterans with PTSD. A quasi-experimental pilot study examined veterans diagnosed with PTSD who completed a yoga intervention in gender-specific groups [69]. Improvements between baseline and

postintervention were statistically significant for PTSD symptoms as well as for depression, sleep, quality of life, and subjective neurocognitive complaints.

Another single-arm pilot yoga intervention designed for veterans with diagnosed comorbid chronic pain and PTSD was conducted at a large urban Veterans Affairs Medical Center [70]. The sample was primarily African American (69%) and male (61%), with a mean age of 51.41 years. The results indicated reductions in overall PTSD symptoms. Veterans reported significant improvements in their ability to participate in social activities and significant reductions in kinesiophobia, an especially helpful improvement for treating pain.

A recent study randomly assigned participants (91.4% veterans; 66% male; 61.7% White; 75.1% currently on psychiatric medication) who met diagnostic criteria for PTSD to attend one of two weekly interventions, yoga or a wellness lifestyle program, for 16 weeks [71]. Participants in both groups showed substantial decreases in PTSD severity over the course of treatment, although those in the yoga group showed significantly greater reductions in PTSD severity at treatment end than did those in the lifestyle program. Group differences persisted at 7 month follow-up but were no longer statistically significantly different.

Several studies have focused on community residents self-reporting high levels of PTSD symptoms. One clinical trial recruited people from the community who had PTSD symptoms, with eligibility based on reporting clinically significant levels [72]. Approximately 60% of the sample reported having been diagnosed with PTSD and 47% reported taking psychiatric medication. Participants were randomly assigned to eight weeks of Kundalini yoga or a waitlist. Participants in both groups demonstrated decreases in PTSD symptoms, although those in the yoga group reported a significantly greater decline. Participants in the yoga group also showed greater improvements in measures of sleep, positive affect, perceived stress, anxiety, stress, and resilience relative to those on the waitlist.

A community sample of 38 women (52% White, mean age of 44, 25% veterans) with self-reported full or subthreshold PTSD symptoms was recruited and randomized to a 12-week Kripalu yoga intervention or a waitlist [73]. Yoga participants reported fewer reexperiencing and hyperarousal symptoms by the end of the intervention. The waitlist control group, however, showed similar decreases in reexperiencing and anxiety symptoms, which may be a result of the positive effect of self-monitoring on PTSD and associated symptoms or may indicate regression to the mean.

Finally, a clinical trial in Colombia randomized ex-combatants from illegal armed groups who had been diagnosed with PTSD (73% male, 9% reported prior treatment) to either sixteen weeks of twice-weekly yoga or a waitlist control [74]. Both groups decreased in their levels of PTSD symptomatology over the sixteen weeks, although the yoga group reported a 19% greater reduction in symptoms.

*6.2. Summary of Yoga for PTSD*

Although preliminary, these studies collectively provide promising support for the use of yoga as an adjunct to pharmacological or psychotherapeutic approaches to treating PTSD. The effects are substantial and are in addition to any effects observed for other treatments participants may have been receiving.

Adding yoga practice to standard treatments for PTSD is based on a solid theoretical rationale. Although the specific mechanisms by which yoga improves outcomes for people with PTSD have yet to be empirically established, several theories have been put forward. Yoga, with its combination of controlled breathing, relaxation, meditation, and movement, can shift autonomic balance towards the parasympathetic branch of the autonomic nervous system, thereby reducing the hyperactivation of the amygdala and elevated cortisol levels that often accompany PTSD [68]. Yoga can alleviate PTSD via psychological pathways as well. As noted above, yoga can promote better emotion regulation, helping individuals tolerate and persist in psychotherapy [35], and may increase mindfulness, which helps reduce the avoidance that is characteristic of PTSD [75].

## 7. Eating Disorders

Eating disorders are categorized into three main types: anorexia nervosa, bulimia nervosa, and binge eating disorder. The 12 month prevalence rates of anorexia nervosa and bulimia nervosa are, respectively, 0.4% and 1–1.5% of the population, with 10 times as many females affected as males for both disorders [11]. For binge eating disorder, gender ratios are much less skewed, with 12 month prevalence rates of 1.6% for women and 0.8% for men [11]. The reported relapse rates for anorexia nervosa range between 9% and 52%, with most studies reporting rates greater than 25% [76]. Although 45% of people with bulimia nervosa were reported to be in remission and 27% improved at follow-up, 23% had a chronic course [77]. The mortality rates are relatively high for anorexia nervosa (4.0%) and bulimia nervosa (3.9%) and are further elevated by high rates of suicide [78]. Comorbidity with anxiety disorders and depression is common [79].

Yoga is frequently used in a multimodal treatment for people with eating disorders [80]. Yoga can address the comorbid depression and anxiety associated with eating disorders, as well as body dissatisfaction and negative affect [81,82]. It also, in theory, addresses a weakness of other cognitive therapies, as it focuses less on the cognitive aspects of the disorder (e.g., judging) and more on accepting and being present in the body [83].

### 7.1. Yoga as Prevention of Eating Disorders

Young adults in a population-based survey who engaged in yoga and Pilates were not at lower risk of engaging in disordered eating than those who did not [84]. Young women who reported participating in yoga or Pilates were less likely to report body dissatisfaction than those who did not, but they did not differ significantly in their weight control methods and binge eating. On the other hand, young men were more likely to use extreme strategies for controlling their weight and to binge eat.

Some studies examined changes in eating disorder symptoms and body satisfaction in general community samples, focusing on preventing or reducing symptoms of eating disorders. For example, in a naturalistic design, fifth grade girls self-selected either a yoga intervention (*N* = 91, 14 weeks of 90 min yoga plus prevention curriculum) or a waitlist control (*N* = 41) [83]. Over the course of the program, girls in the yoga intervention group significantly decreased in their drive for thinness and body dissatisfaction and increased their self-care at the post-test when compared to girls in the control group. This study partially replicated an earlier study in a similar but somewhat shorter intervention without a control group [85]. That study reported decreased body dissatisfaction, decreased tendencies to think about and engage in uncontrolled eating, and improved social self-concept.

In another study, college students (*N* = 99, 77.8% female) completed a yoga class that met for 50 min, three times per week, for eight weeks [86]. Relative to the beginning of the course, body dissatisfaction and eating disorder pathology were significantly lower at the end of the course, while body appreciation and self-compassion increased. Men reported fewer concerns about being overweight and more improvements in body satisfaction as a result of the course than did women. Finally, 113 women were randomly assigned to a yoga intervention, a cognitive dissonance intervention, or a control group; 93 women completed the post-intervention assessment [87]. Information about psychotherapy or medication use was not provided. Both interventions met weekly for 45 min/week over six weeks. Pre-to-post intervention, women in the cognitive dissonance group demonstrated significant decreases in disordered eating, drive for thinness, body dissatisfaction, and alexithymia. No significant differences over time were reported for the yoga or waitlist group. No changes in anxiety or depression scores were reported for any group.

Other studies focused on at-risk adult populations in the community. Ninety overweight or obese participants from a community sample with a binge eating problem were randomly assigned to a weekly 60 min hatha yoga session for 12 weeks or a waitlist control [88]. The yoga group's binge eating scores and physical activity significantly decreased, while the waitlist controls showed no changes. These differences were maintained at a

three month follow-up. Five women with a history of both eating disorders and doing yoga completed a six day Forrest yoga workshop (yoga, healthy cooking, and reflection) with follow-ups one month after the workshop [89]. Over the course of the study, the participants' ability to recognize and respond to emotional states and affective problems improved. They also reported decreases in eating disorder symptoms and psychological maladjustment between baseline and postintervention, effects that were maintained at the one month follow-up. Finally, a group of adult women (*N* = 52, 25–44 years old) who were classified as restrained eaters with reported elevated stress were randomly assigned to either a Bikram yoga intervention (two 90-min classes weekly for eight weeks) or a waitlist control [90]. No participants were also receiving concurrent mental health treatment of any kind. Yoga participants reported greater decreases in the frequency with which they were binge eating and less eating to cope with negative emotions.

### 7.2. Yoga as a Complementary or Integrative Therapy for Eating Disorders and Disordered Eating

In a single-arm study, 20 adolescents with mixed eating disorder diagnoses (75% diagnosed with "other specified feeding or eating disorder") enrolled in a 12 week trial of hatha yoga (between 60 and 90 min per week), although only 70% completed the baseline assessment [91]. Of the total enrolled, 45% had prior experience with yoga; 55% completed at least seven of the 12 sessions and the final assessment at 12 weeks. No information on the nature of other treatments received was reported. Participants reported no changes in restraint and eating concerns but significant decreases in weight concern, shape concern, anxiety, and depression.

Norwegian adults who were solicited from sites with access to people with eating disorders were randomly assigned to either a hatha yoga group (twice weekly for 90 min) or a waitlist control [92]. Twenty-one completed assessments at the post-test, with 12 of 18 and 9 of 12 participants, respectively. Six were also receiving concurrent psychotherapy—four in the yoga group, two in the control group. No information about medication was reported. Scores on the global, restriction, eating concern, and weight concern scales of the Eating Disorders Examination decreased across time for the yoga group but not the waitlist control group. No changes in shape concern were observed. Changes were maintained at six month follow-ups. In another trial, adults meeting criteria for either bulimia nervosa or binge eating disorder were randomly assigned to either a Kripalu yoga (90 min weekly for 8 weeks; *N* = 26 completers) or a waitlist control group (*N* = 27 completers) [93]. Many had a history of prior psychotherapy (79.2%) and current medication (30.19%). Relative to waitlist controls, participants in the yoga group showed greater decreases in binge eating frequency, emotional regulation difficulties, and self-criticism and improved self-compassion and mindfulness skills over the course of the study.

Adolescents receiving outpatient treatment for an eating disorder were randomly assigned to either a TAU waitlist or an eight-week trial of individual yoga (twice weekly for an hour) plus TAU [81]. Participants were female (92%), diagnosed with anorexia nervosa (55%), and had a history of hospitalization (45%) and overexercising (48%). At the end of treatment, adolescents in the yoga group reported fewer symptoms of eating disorders, which was maintained at a one month follow-up. Symptoms of depression and anxiety declined across time for both groups, which were not significantly different from each other. Food preoccupation declined between the start and end of each yoga session.

Thirty-eight participants in a residential eating disorders program (97.3% female, 58% diagnosed with anorexia nervosa) were randomly assigned either to an hour-long hatha yoga class for five days immediately before dinner or a control group completing regularly scheduled residential activities during the same period [82]. Pre-meal but not post-meal negative affect declined among participants in the yoga group.

### 7.3. Summary of Yoga for Eating Disorders and Disordered Eating

Yoga appears to be a promising adjunct for treating people with eating disorders and may be an efficacious prevention strategy in reducing risk factors and symptoms when

used in the community (e.g., [83,86,88]). Although the studies reviewed generally drew positive conclusions about yoga's efficacy with these populations and make sense from a theoretical standpoint, the research evidence is weak, as many studies lacked a control group and studies that did employ a control group often used waitlist controls rather than active control groups. In a meta-analysis, however, effect sizes for global eating disorder psychopathology and body image concerns were small but significant, and for binge eating and bulimia were moderate to large [6]. Effect sizes for dietary restraint and eating concerns were not significant.

## 8. Conclusions

Any general conclusions drawn from our review must be tempered by acknowledging the lack of rigor of this body of research, and we discuss in some detail these limitations below; however, the extant studies have fairly consistently yielded findings suggesting that yoga may be a helpful addition to first-line therapies for people diagnosed with anxiety, depression, and other psychiatric disorders. Yoga may be an appealing option for many people who are hesitant about psychiatric medications or psychotherapy. Yoga is widely available and can be made affordable [6]. Further, although occasional injuries were described in the research reported here, yoga has few negative side effects, which are mostly musculoskeletal issues such as strains and sprains [3]; nonetheless, some studies of people with eating disorders reported problems with social comparisons or negative self-talk [6].

Many plausible mechanisms by which yoga may affect symptoms have been identified, although their function as the link between yoga practice and symptom reduction remains to be tested. Importantly, many of these mechanisms are transdiagnostic. For example, stress is known to exacerbate virtually every mental health symptom [94] and yoga's stress-reducing properties have been well-documented [3]. Yoga helps to shift autonomic balance towards the parasympathetic nervous system, which provides a cascade of benefits (increased frequency of heart rate variability, decreased GABA), manifesting as reductions in many types of symptoms [3,9,33,36]. Stress reduction through yoga also provides myriad psychological benefits that may reduce mental health symptoms, including increased emotion regulation capability, improved health behaviors, better social connections and support, and deeper spirituality [95].

The current review was based on studies identified in previous systematic reviews along with additional literature searches, but does not constitute a full systematic review of yoga for each disorder. In addition, the review is based on a body of literature that is generally not strong; however, the current limitations of this body of literature provide ample direction for more rigorous research in the future. Many of the reviewed studies relied on treatment as usual or waitlist controls, which did not account for non-specific factors such as expectancies and attention. Future well-designed clinical trials should randomize participants into conditions of yoga and comparators, so that the added value of yoga can be determined. Many of the studies we reviewed did not even report whether participants were also receiving medication or psychotherapy let alone control for them in analyses. We know little about the therapeutic "dose" of yoga, and most of the trials of yoga reported here were very short; yoga's beneficial effects likely accumulate over months and years.

Hatha yoga is a wide umbrella of many distinct types of yoga that vary on many dimensions. Studies should carefully select and tailor the yoga intervention to the targeted symptoms, perhaps considering the specific mechanisms targeted. For example, yoga that emphasizes breathwork and linking breath to movement may be particularly helpful in improving vagal tone, while more gentle or restorative types of yoga may target the relaxation response. More vigorous yoga may help patients to improve their cardiovascular system and muscular strength, which may improve emotion regulation abilities [96]. To date, however, different yoga styles do not appear to differ in their rates of positive outcomes [4].

In summary, yoga appears to be a viable complementary intervention approach for a range of mental health challenges. The research evidence in support of its beneficial effects is not yet strong but is becoming more robust over the years as researchers implement more rigorous trials. It is appealing to many people and of low risk and low cost. Further, there are many different types of yoga, meaning that people have options regarding different styles and intensities and may be able to find the best fit for developing a yoga lifestyle that helps them achieve and maintain better mental health.

**Author Contributions:** Conceptualization, writing of both the original draft and reviewing and editing, C.L.P. and J.M.S. All authors have read and agreed to the published version of the manuscript.

**Funding:** This research received no external funding.

**Conflicts of Interest:** The authors declare no conflict of interest, financial or otherwise.

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
