# Peer review of "Yoga as an Integrative Therapy for Mental Health Concerns: An Overview of Current Research Evidence"

_2673-5318, doi:10.3390/psychiatryint2040030_

Round 1
Reviewer 1 Report
The authors made all the suggested corrections and changes.
I have only couple comments:
- there is a double numbering in the references
- in the sentence "In the US, for example,approximately 13.2% of adults report having practiced yoga [4], almost 170% more in 2020 45 than in 2012 [4]. (lines 44-6) there is no need to double cited which is the same - [4]. You may leave it only at the end of the sentence
Author Response
Thank you for your help in revising the paper. In response to your recent comments, we have corrected the numbering in the paper.
For the citation that we had double-cited, we have removed one of the citations.
Reviewer 2 Report
Dear Author,
Good afternoon. I think with the revision, the submission is good.
Thanks for the opportunity to review the revised submission.
Author Response
Thank you for your help with the revision. We appreciate your efforts!
Reviewer 3 Report
I thank the authors for their revision of the manuscript. I think that my concerns have now adequately been met and that the wording of the conclusions now matches the methods use to derive them.
Author Response
Thank you for your help with the previous revisions. We agree the paper is stronger and clearer now.